



# Modelling the potable water quality in a distribution network based on the hydraulic conditions

Jani Tomperi[1]

[1]Control Engineering, Environmental and Chemical Engineering, University of Oulu. Oulu, Finland.
P.O. Box 4300, FI-90014 University of Oulu, Finland.

*Correspondence to*: Jani Tomperi (jani.tomperi@oulu.fi)

**Abstract.** Abnormalities in hydraulic conditions inside a water distribution network are strongly related to the deterioration of potable water quality. Leaking pipes and valves, for instance, cause changes in water hydraulic conditions and may allow the entry of microbes to the distribution system. Flow and pressure shocks can detach soft deposits and biofilms from the

pipe surface which is shown among others as the elevated concentrations of bacteria, metals and turbidity in water. On that account, monitoring the hydraulic conditions in a distribution network and utilizing this information in developing a predictive water quality model assists providing a sufficient amount of potable water with an appropriate quality for the consumers use. In this paper, the water quality at the end part of the district metered area is modelled based on only the water flow and pressure measurements along the distribution network. The developed model can be utilized in proactive operation

as it is able to show the potable water quality hours in advance before it is discovered at the end part of the distribution network.

## 1 Introduction

Providing potable water with appropriate quality and quantity through the water supply system - the infrastructure that collects, treats, stores, and distributes water between raw water sources and consumers' taps - is essential not only to ensure

compliance with water quality standards and guidelines but also for the general well-being and health of consumers. In addition to potential health-related risks, extra financial costs and energy consumption occur when potable water is contaminated. The deterioration of potable water quality strongly relates to problems in a water treatment process or abnormalities and sudden changes of the hydraulic conditions inside a water distribution network (WDN) (Clark & Haught, 2005).

The WDN is one of the significant contamination sources of potable water as microbiological growth, resuspension and mobilization of precipitation, or breakdown of pipes and valves allowing the entry of microbes, can deteriorate the water quality. When distributing drinking water, it is inevitable that biofilms will grow on the inner surfaces of the pipes and soft deposits consisting of organic and inorganic matter and several metals will accumulate to the pipelines. Rapid changes in water flow or pressure can detach biofilms and soft deposits from the surfaces of pipes and deteriorate the water quality





which is shown as the elevated concentrations of bacteria, metals and turbidity in water, among others (Lehtola et al., 2006b). The rapid change in the flow creates high shear stresses which cause particle mobilization from sediments and biofilms along the pipe affecting the water quality. Pressure shocks increase temporarily the number of particles, turbidity and electrical conductivity in water (Mustonen et al., 2008). In addition, unsteady hydraulic conditions can also lead to significant impact on the disinfectant residual (Aisopou et al., 2012). A significant decay in the disinfectant residual is

attributed to the mobilization and entrainment of particles and biofilms which affect the bulk and wall reactions during the unsteady hydraulic conditions. Pump failures, pipe breaks or sudden large increases in local water demand decrease the pressure and water flow inside the pipeline. The lack of pressure and low velocity leads to long detention times that may increase the biofilm growth and bacterial regrowth which may ultimately lead to water-borne diseases as the long travel and detention times contribute to the loss of disinfection residual. Long detention times are a significant contributing factor in the

formation of disinfection by-products (DBPs) which are formed when the disinfectant reacts with organic and inorganic substances in water. Many DBPs have toxic properties and can be mutagenic and genotoxic. (Clark and Haught, 2005; Ghebremichael et al., 2008; Mains, 2008; Li, 2017; Manasfi, 2017)

The water quality deterioration and the events causing it may be rapid and occur over a short time period. The changes in water quality typically have characteristics of a sharp rise that reduces within a few hours (Vreeburg and Boxall, 2007).

Typically, the largest waterworks are required to monitor and report the quality of the potable water only once a day, and the smaller waterworks may monitor and report it even less frequently. The quality monitoring is often carried out by manual sampling and laboratory analysis. Therefore, the monitoring frequency is not sufficient to detect the abrupt and transient events in real-time, but the deterioration of water quality is noticed by the customers. Sampling and analysis of the water after customers complains does not reveal the quality of transiently deteriorated water or reasons behind it when done hours

afterwards. Hence, monitoring the water distribution network to reveal the changes in hydraulic conditions in real-time and utilizing this information on assessing the water quality is an issue of a great interest. Water distribution networks generally include at least some amount of basic on-line pressure and flow measurements that can reveal the abnormal situations inside the distribution network. These measurements show the changes in hydraulic conditions in real-time and this information can be utilized to indicate the upcoming problems and changes in the water quality with a developed and implemented predictive

water quality model.

Water quality models have been developed to help ascertain the quality of potable water in the water distribution systems and for design and operational management purposes. For example, an integrated pressure-dependent hydraulic model based on the well-known Epanet 2 model was used to simulate the operating condition including normal and subnormal pressure (Seyoum and Tanyimboh, 2013). On the other hand, a data-driven turbidity forecasting method capable of aiding operational

staff and enabling proactive management strategies was developed by Meyers et al. (2017). Development of a hydraulic model includes many challenges as it require detailed information on the distribution network and may therefore include some assumptions which cause uncertainties to the model performance. Significant spatial and temporal variations in hydraulic conditions can occur inside the water distribution pipeline. While the pressure is nearly constant in stable

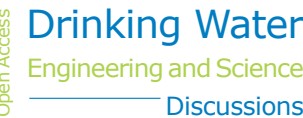

conditions, the water flow rate varies within a day, from day to day and season to season. The dynamic behaviour poses
challenges as the water usage is time-dependent and tied to the type of the water users, for example residential or industrial.
In addition, the development and maintenance of hydraulic models are time-consuming and expensive. Data-based models,
on the other hand, require only the measured data on the hydraulic conditions and water quality. However, the accuracy of
the data-based model is strongly dependent on the quality and quantity of measured data.

In this paper, a water turbidity forecasting model developed for an early warning tool is presented. The quality of potable
water in the end part of the real district metered area (DMA) is modelled based on the on-line hydraulic condition
measurements along the WDN. Currently, disturbances in potable water quality caused by changes in hydraulic conditions
are most commonly handled with a reactive way by water companies, i.e. expensive cleaning parts of the distribution
network (water flushing, water/air scouring and swabbing/pigging) (Vreeburg and Boxall, 2007) and providing clean water
to consumers using tanker trucks etc. With a help of the predictive water quality model, if abnormal situation occurs,
proactive management strategies can be utilized by operation personnel and hence prevent the poor-quality water from
ending up to consumers, and also reduce the operational costs and excessive energy consumption. In this research, the
developed turbidity model is completely data-based and therefore does not require hydraulic or network models that are
expensive to develop and maintain. On-line measured data of water quality and hydraulic conditions collected from a real
WDN during a period of a half year is used to develop and verify the model.

## 2 Material and Methods

### 2.1. The research site and collected data

The research site was a real DMA in eastern Finland, where on-line monitoring stations were installed along the water
distribution system (Figure 1) including a raw water source pumping station (marked as I in Figure 1), pressure increase
pumping stations along the DMA, and distribution pipelines at the length of tens of kilometres. The complete data used in
this study consist of the continuous on-line water quality and hydraulic conditions measurements from a period of over six
months from early spring to late fall. Hydraulic condition parameters were monitored in several locations (marked as circles
in Figure 1). Some of these measurement locations also included parallel flow and pressure sensors for the comparison of
different sensors. Originally at a minute interval logged data were hour-averaged for this research. The water quality
measurements were performed at two locations: at a raw water pumping station and at one monitoring station at the end part
of the distribution network (marked as O in Figure 1). Water quality measurements were performed with YSI 6920V2, multi-
parameter probes for continuous water quality monitoring, which included in addition to turbidity sensor dissolved oxygen,
oxygen reduction potential, pH, salinity, specific conductivity and temperature sensors (Xylem, 2019).





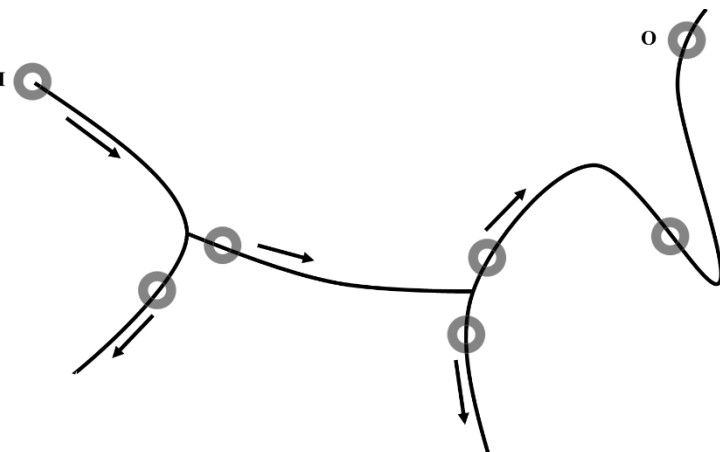

**Figure 1: A rough illustration of the WDN and the measurement locations.**


Properties that affect the water quality can be physical (e.g. temperature, turbidity), chemical (e.g. pH, dissolved oxygen) or biological (e.g. algae, phytoplankton) factors. Turbidity is one of the key water quality parameters in environmental monitoring, industrial process operation and water treatment and distribution. Turbidity is an optical measure of clarity describing the physical transparency of liquid. High turbidity water is cloudy or muddy while water with low turbidity is

clear. Turbidity and total suspended solids are strongly related but turbidity is not a direct measurement of the total suspended materials in water. In a water distribution system suspended particles that cause the water turbidity are usually from raw water source due to inadequate water treatment or from the re-suspension of sediments in a distribution system. Turbidity and water flow are causally related as the high flow prevents particles from settling and large changes in velocity can increase turbidity and corrosion in the distribution system (Chapman, 1996). The increase in turbidity can often indicate

potential pollution as pollutants such as dissolved metals and pathogens can attach to suspended particles. It has been found that the number of bacteria in water correlates with turbidity and the number of particles in water (Lehtola et al., 2006a). Despite its unpleasant appearance, turbid water by itself is not harmful to health, but for the potable water, maximum value of 5 NTU has been set by WHO because a high turbidity of water hinders the effect of disinfection. The suspended particles can carry impurities, protect microorganisms from the effects of disinfection and stimulate bacterial growth. Hence, the

effective disinfection requires as low as 0.1 NTU turbidity. (Chapman, 1996; Anderson, 2005; WHO, 2008; Mukundan et al., 2013)

**2.2. Data analysis and modelling**

The data collected from the on-line sensors were combined and scaled between 0 and 1 before modelling to treat the variables equally and to ensure that the statistical distribution of variables is roughly uniform. Auto-correlation analysis



measures the similarity between two variables as a function of the lag and was here used to indicate the delays between measurement locations. (MathWorks, 2019)

Datasets generally include irrelevant variables to develop a model for a specific variable. Selecting the proper amount and the optimal input variables for a model from a large variable set is one of the most important steps in model development and can be done manually or using various automated variable selection methods. Too few input variables or using noisy and

uninformative variables lead to a model with a poor performance. On the other hand, using too many input variables increases the risk of developing an over-fitted model with excellent training results but poor prediction abilities with new data. In this study, stepwise regression was used as a preselection method to find the optimal input variables for modelling the water quality parameters. Final selection of input variables was made based on expert knowledge. Stepwise regression is a modified forward selection method, which adds the best variable to, or deletes the worst variable from a variable subset at

each round. Adding and deleting is based on variable's statistical significance in regression. It starts with an initial model and continues until either no further model changes occur over one complete round or a pre-set number of variable selections and deletions occur. (MathWorks, 2017)

As the available data was relatively short, around six months, it was not feasible to use an external dataset for model validation but instead, a cross-validation was used. A cross-validation is an efficient resampling method to predict the fit of a

model for a validation set when dataset is small and an explicit validation set is not available. In cross-validation, the whole data set is used for training and validating the model by using part of the data for training and the rest of the data for validation and repeating this until the whole dataset is processed. A single estimation is then produced by combining these results.  Thus, the largest possible subset is used for the both training and validation of the model. (Arlot and Celisse, 2010)

In this study, multivariable linear regression (MLR) was used to predict the output variables as a linear combination of

selected input variables as in Eq. (1):

$$y = b + p_1x_1 + p_2x_2 \cdots + p_nx_n \qquad , \tag{1}$$

where y is the predicted output variable, xs are the selected input variables, and ps and b are model parameters and a bias value defined from the data. Although, the linear model does not find the nonlinear relations between input and output variables, but it has a simple structure and is easy to understand and to implement. MLR models are also suggested to be

used to avoid over-fitting and in some cases they can outperform the more complex and computationally heavier nonlinear models. Root Mean Square Error (RMSE) and coefficient of determination ($R^2$) are used to evaluate the relative performance of the model. (Hastie et al., 2009; Montgomery et al., 2012)





## 3 Results and Discussion

The delays between the monitoring locations in the DMA were studied both mathematically using auto-correlation analysis and inspecting visually the flow, pressure and quality measurements figures to compare the temporal location of the peaks and level changes. Both mathematical and visual inspections showed that variables from the consecutive monitoring stations in the WDN have logical delays from a few hours up to tens of hours. Based on the mathematical analysis, the quality measurements in the end part of the network have a delay up to over 20 hours to the pressure and flow changes in the

beginning of the distribution network.

In Table 1 is shown the performance values $R^2$ (in which 1.0 means the perfect match) and RMSE of the developed water turbidity model together with the selected input variables ($x_n$), and the bias (b) and the parameter ($p_n$) values of the model. When looking at the model validation results, it is important to take into consideration that the model only uses flow (F) and pressure (P) measurements as inputs, but no water quality measurements at any point of the distribution network. For

example, the temperature is an important factor to take into account when assessing the water quality as the temperature can alter the physical and chemical properties of water and influences several quality parameters. However, the performance of the developed turbidity model is good which indicates that the turbidity at the end part of this real WDN can be estimated based on only the water hydraulic conditions monitoring. As seen, the turbidity is highly dependent on the measured pressures and only one flow measurement has a strong influence on the turbidity. Presumably pressure shocks increase the

number of particles in water which is shown as increased turbidity. The developed model could be implemented into the automation system of the water treatment plant and used as an early warning tool to indicate the upcoming changes in water quality. With this information, the personnel could operate more proactively nor reactively, and thus avoid excessive operation cost and achieve the distribution of potable water with required quality.

**Table 1. The performance values, selected input variables and the parameter values of the developed water turbidity model.**

|  | Turbidity model |
| --- | --- |
| $R^2$ | 0.77 |
| RMSE | 0.18 |
| $b + p_1*x_1 + \ldots + p_n*x_n$ | $-2.13 + 2.15*F_1 + 2.91*P_2 + 0.33*P_3 - 0.42*P_4 - 0.33*P_5$ |

## 4 Conclusion

In this study, the flow and pressure measurements of the real distribution network were used as input variables to develop a water quality model to predict the turbidity in the end part of the distribution network. The on-line monitoring data were

collected at several monitoring stations along the urban water distribution network during a period of over six months.



Based on the modelling results, turbidity can be assessed by the flow and pressure measurements of the distribution network. When looking at the model results it is important to take into consideration that the model only uses flow and turbidity measurements as inputs, thus any turbidity caused by other factors may not be picked up. As the data analysis also showed, the delays between the monitoring stations and the water quality monitoring point were several hours and the information

received from the model can be used as an early warning of the changes in water quality. This enables proactive operation strategies by the personnel and hence prevents the poor-quality water from ending up to consumers, and in addition reduce the operational costs and excessive energy consumption.

**Acknowledgements**

This research was carried out as part of the ITEA2 Water-M project. The data used in this research is the property of Kuopion Vesi and it was downloaded from SaMi, the Savonia Measurements site, with permission.

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
