# Peer review of "Modelling the potable water quality in a distribution network based on the hydraulic conditions"

_Drinking Water Engineering and Science, 2020_

## Referee Comment (RC1) · Anonymous Referee #1 · 6 Sep 2020

The manuscript claims to present a method to predict water quality solely by applying a data-based multilinear regression model on flow and pressure measurements of a real drinking water distribution system. It does not contain any data or figures that indicate that the method works. It is even hard to judge if the case study area exists, since the only figure is a rather abstract sketch of the measurement locations, nor if it was applied on data from a real distribution network. Furthermore, the methodology is not described in such a way that allows to repeat the experiments on other case study areas. From my perspective, it is even questionable, if a multilinear regression model is suitable for the tasks claimed in this paper, e.g., the author does not mention how he would treat the multicollinearity problem arising from highly correlated pressure and flow measurements. In conclusion, the reviewer could not recommend the manuscript

for publication in DWES since it does not fulfil scientific standards at all.

---

## Referee Comment (RC2) · Anonymous Referee #2 · 11 Sep 2020

In the manuscript the idea of a data-based turbidity model is presented. However, no actual data and results, apart from a single table with statistical parameters, are included in the manuscript. Methodology section is not written in a clear way, and therefore needs to be improved. In addition to this, less general comments should be made and specific comparisons with literature should be performed in the discussion section. Overall: Even though the idea of the paper is interesting, the author needs to rewrite the paper in order to meet the DWES Journal standard